# Entropic Confinement and Mode Connectivity in Overparameterized Neural Networks

**Luca Di Carlo**[*]
Joseph Henry Laboratories
Lewis-Sigler Institute
Center for the Physics of Biological Function
Princeton University
Princeton, NJ, USA
`lucadc@princeton.edu`

**Chase Goddard**[*]
Joseph Henry Laboratories
Princeton University
Princeton, NJ, USA

**David J. Schwab**
Initiative for the Theoretical Sciences
The Graduate Center, CUNY
New York, NY, USA

## Abstract

Modern neural networks exhibit a striking property: basins of attraction in the loss landscape are often connected by low-loss paths, yet optimization dynamics generally remain confined to a single convex basin (Baity-Jesi et al., 2019; Juneja et al., 2023) and rarely explore intermediate points. We resolve this paradox by identifying entropic barriers arising from the interplay between curvature variations along these paths and noise in optimization dynamics. Empirically, we find that curvature systematically rises away from minima, producing effective forces that bias noisy dynamics back toward the endpoints — even when the loss remains nearly flat. These barriers persist longer than energetic barriers, shaping the late-time localization of solutions in parameter space. Our results highlight the role of curvature-induced entropic forces in governing both connectivity and confinement in deep learning landscapes.

## 1 Introduction

Deep neural networks trained, in the overparametrized regime, exhibit a number of surprising and counterintuitive properties. One of the most striking is the observation that distinct solutions, found with standard optimization algorithms, are often connected by low-loss paths in parameter space (Garipov et al., 2018; Draxler et al., 2018; Frankle et al., 2020). Such *mode connectivity* results imply that the landscape is far less rugged than once assumed: minima that appear isolated are, in fact, linked by paths of low, nearly constant loss. At the same time, however, optimization dynamics display a seemingly contradictory behavior. Standard training with stochastic gradient descent (SGD), with or without momentum, converges to a well-defined minimum and rarely explores regions of parameter space corresponding to these paths (Baity-Jesi et al., 2019).

We argue that this paradox can be resolved by recognizing the role of *entropic forces* generated by curvature variations along connecting paths. Although the loss may be nearly flat along these paths, the curvature of the landscape typically increases away from found minima, producing effective barriers that bias stochastic dynamics back toward the endpoints. These barriers emerge from the interaction between fluctuations induced by SGD noise and the Hessian spectrum along low-energy paths. In this way, regions of parameter space that are energetically connected become effectively disconnected.

---

[*]Equal Contribution

## 1.1 RELATED WORK

Our work aims to unify insights from two areas: First, we draw on insights from a body of work that shows SGD has an *implicit bias* towards flatter minima the strength of which increases with smaller minibatch size.

This behavior is attributed to the higher noise levels in gradient estimates, which act as a form of implicit regularization preventing convergence to sharp minima and instead favoring wider, flatter basins that generalize better (Keskar et al., 2017). Several works have formalized this intuition: Jastrzebski et al. (2018) and Smith & Le (2018) interpret the minibatch noise as an effective temperature that enables exploration and escape from sharp valleys; Wei & Schwab (2019) and Xie et al. (2021) further show that the resulting dynamics can be modeled as a stochastic process biased toward flat regions. Collectively, these studies demonstrate that curvature and generalization are deeply intertwined with the stochastic geometry of the optimization trajectory. We leverage these insights to conduct a deeper analysis of *mode connectivity* in neural network training. Garipov et al. (2018) and Draxler et al. (2018) showed the existence of nonlinear paths of low-loss between different minima found by training with different random seeds. Subsequently, Frankle et al. (2020) showed that when the training dynamics of two networks are tied together early in training, the resulting minima, found after training is complete, are *linearly* connected by paths of low-loss. Follow-up work has analyzed in more depth how linear mode connectivity emerges (Singh et al., 2024; Zhou et al., 2023), and such work has important implications for model merging (Ainsworth et al., 2023; Singh & Jaggi, 2020) and weight-space ensembling (Izmailov et al., 2018; Wortsman et al., 2021; Gagnon-Audet et al., 2023; Wortsman et al., 2022).

## 1.2 CONTRIBUTIONS

Our main contributions are as follows:

- We show empirically that the curvature along minimum-loss paths between minima increases away from the endpoints.
- We argue that such a "bump" in the curvature leads to an *entropic barrier*, and that such entropic barriers lead to confinement of solutions even when the loss is low along a path in parameter space.
- We show that despite the existence of low-loss connecting paths between solutions, entropic forces confine models to specific regions of parameter space.
- We show that entropic barriers between minima persist longer than energetic barriers, when considering models that shared the first $k$ epochs of training, suggesting that both energetic *and* entropic forces are responsible for the final region of parameter space that a model ends up in.

## 2 BACKGROUND: ENTROPIC FORCES AND CURVATURE

It is well established in statistical physics that the state of a system is governed not only by energetic forces—derived from gradients of an energy or potential—but also by entropic forces, arising from thermal fluctuations. In the context of neural networks, the energy landscape is defined by the training loss, and its gradient directs the deterministic component of learning. However, the stochasticity introduced by finite learning rates and minibatch sampling induces an effective temperature, making entropic contributions to the trajectory of the model non-negligible. As a result, optimization can be biased toward broader, flatter regions of the landscape—not because they are lower in loss, but because they occupy a larger volume in parameter space. We illustrate this idea in a simple toy model. Consider a Brownian particle evolving in a two-dimensional potential

$$\dot{\boldsymbol{x}} = -\nabla V(\boldsymbol{x}) + \boldsymbol{\xi}(t), \qquad V(x, y) = \frac{1}{2} g(y) x^2, \qquad g(y) > 0 \tag{1}$$

where $\boldsymbol{\xi}$ is white delta correlated Gaussian noise, $\langle \xi_i(t) \xi_j(t') \rangle = 2T \delta_{ij} \delta(t - t')$, and $g(y)$ is an arbitrary positive function of the coordinate $y$. In the analogy to deep learning, $V$ plays the role of the task loss function, while the noise $\xi$ arises from SGD noise due to minibatching and finite learning rate. To leading order one can identify $T \propto \eta/B$, for a learning rate $\eta$ and batch size $B$,

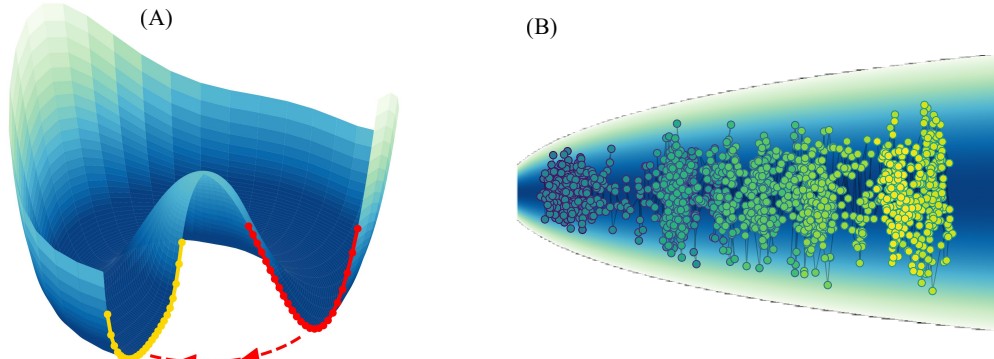

Figure 1: **Curvature produces an entropic force.** (A) Illustration of a potential $V(r, \theta)$ with a circular minimum at $r = 1$, where the curvature varies with angle. At zero temperature ($T = 0$), the angular distribution is uniform, $P(\theta) = 1/(2\pi)$. At finite temperature, thermal fluctuations bias the system toward flatter regions (yellow) rather than sharper ones (red). (B) Example of a Brownian particle diffusing along the ridge of a loss landscape, lighter colors correspond to larger times. Entropic forces generated by fluctuations push the particle toward flatter directions, effectively favoring broader regions of the landscape.

(Mandt et al., 2017; Smith et al., 2020; Liu et al., 2021), though the precise relationship depends on local curvature and other details of the loss landscape (Ziyin et al., 2021). In this simple motivating example we assume $\xi$ is white and Gaussian, even though in deep networks the SGD noise is neither perfectly white nor perfectly Gaussian. In this analogy, $y$ corresponds to soft modes (directions with nearly flat curvature where the loss hardly changes), and $x$ represents the stiff modes (directions associated with large eigenvalues of the Hessian).

For fixed $y$, the distribution of $x$ is Gaussian with variance $\langle x^2 \rangle = T g(y)^{-1}$. When $x$ relaxes on a much faster timescale than $y$—for instance when the curvature of the potential along the $x$-direction is much larger than that along the $y$-direction—we may average over the fast variable $x$ to obtain an effective dynamics for $y$:

$$\dot{y} = -\partial_y g(y) \langle x^2 \rangle + \xi = -T \frac{1}{g(y)} \frac{dg(y)}{dy} + \xi. \tag{2}$$

This equation can be rewritten as a gradient-flow dynamics generated by an effective potential $V_{\text{eff}}(y)$:

$$\dot{y} = -\frac{dV_{\text{eff}}(y)}{dy} + \xi, \qquad V_{\text{eff}}(y) = T \ln g(y), \qquad \langle \xi(t) \, \xi(t') \rangle = 2T \, \delta(t - t'). \tag{3}$$

The resulting stochastic dynamics converge to a Boltzmann-like stationary distribution (Gardiner et al., 2004),

$$P(y) \propto \exp\left[-\frac{V_{\text{eff}}(y)}{T}\right]. \tag{4}$$

Equation 3 reveals the key mechanism: the force is proportional to the negative derivative of $g(y)$, effectively driving the system towards smaller values of $g(y)$, corresponding to flatter directions in $x$. We call these forces entropic because they are proportional to the effective temperature $T$ and therefore vanish in the absence of noise. This is a familiar principle in statistical physics, where the thermodynamic state of a system is determined by a competition between minimizing energy and maximizing entropy, with the temperature controlling the relative importance of the two. In Fig. 1, we illustrate this effect with two example potentials whose curvature varies with one of the coordinates, showing how the resulting entropic force pushes the system toward flatter regions.

In deep neural networks, these forces are expected to grow stronger as the effective temperature increases, making them more prominent for large learning rates and small minibatches, as we illustrate in Section 4.1. When entropy outweighs energy, entropic forces can even dominate, potentially

driving optimization to *climb* the loss landscape. An example of this phenomenon in a deep network is also shown in Section 4.1.

This minimal example is far from capturing the full complexity of the dynamics in regions of approximately constant low loss of real deep neural networks. Nevertheless, it illustrates how stochasticity interacts with curvature to favor flatter minima. Even when the training loss is near zero, these entropic barriers can effectively confine solutions to specific regions of parameter space. In the remainder of this paper, we show empirically that entropic forces arising from the curvature of loss landscapes in real networks trained on natural images produce qualitatively similar behavior.

## 3 METHODS

We begin by training a collection of image-classification models on CIFAR-10 using both Wide ResNet and ResNet architectures, each initialized from different random seeds to obtain a diverse set of distinct minima. For every pair of minima under study, we construct a low-loss path connecting them using the AutoNEB algorithm of Draxler et al. (2018).

Once the MEP is obtained, we analyze its geometric and dynamical properties. First, we measure curvature along the path using several complementary Hessian-based statistics, allowing us to characterize how flatness varies between the endpoints and the interior of the path. Second, we study optimization dynamics constrained to the MEP by projecting SGD updates onto the nearest path segment. This controlled setting isolates how stochasticity interacts with curvature: models initialized along the MEP exhibit systematic drift toward flatter regions, revealing the action of entropic forces whose strength grows with the effective noise level. We then proceed to analyze this phenomenon at a finer scale by studying linearly connected minima, following the approach of Frankle et al. (2020).

### 3.1 TRAINING DETAILS

Unless otherwise specified, all experiments are conducted on Wide ResNet-16-4 (Zagoruyko & Komodakis, 2016) trained on the CIFAR-10 dataset (Krizhevsky, 2009). Following standard practice (Zagoruyko & Komodakis, 2016), we use stochastic gradient descent (SGD) with momentum $\beta = 0.9$, weight decay regularization $w = 5 \times 10^{-4}$, and an initial learning rate of $\eta = 0.1$. Models are trained for 200 epochs with a batch size of 256, and the learning rate is reduced by a factor of 5 at 30%, 60%, 80%, and 90% of the total training epochs. We apply mild data augmentation consisting of random horizontal flips and random crops with 4-pixel padding followed by resizing to the original $32 \times 32$ resolution.

### 3.2 MINIMUM ENERGY PATHS

To explore the structure of the loss landscape between different solutions, we identify low-loss connecting paths using the *Automatic Nudged Elastic Band* (AutoNEB) algorithm introduced by Draxler et al. (2018). In brief, the algorithm initializes $k$ intermediate *pivots* along the straight-line path between two minima and optimizes their positions such that they evolve as if connected by elastic springs, while minimizing the loss orthogonal to the path.

Since the loss along the straight segments between pivots may still be high, AutoNEB dynamically adds new pivots whenever the loss along a segment exceeds a predefined threshold. This adaptive refinement ensures a smooth, low-loss path is found. Following Draxler et al. (2018), we refer to such paths as minimum energy paths (MEPs), by analogy with physical systems. Importantly, the optimization dynamics is such that it does not change the length of the segments composing the MEP, so when a new pivot is inserted between two existing pivots, the resulting segments remain shorter than the original. In the following, the relative position along the MEP is reported in terms of pivot index, normalized by the total number of pivots. Note that this parameterization does not reflect the actual metric distance along the path, as the pivot density and length are non-uniform, see Figure 7 in Appendix A.3.

Unless otherwise specified, all MEPs shown in the paper are computed using a sequence of refinement cycles with decreasing learning rates. Specifically, we run four cycles each with the following parameters: $(0.1, 10)$, $(5 \times 10^{-2}, 5)$, $(10^{-2}, 5)$, and $(10^{-3}, 5)$, where each tuple denotes (learning rate, number of epochs).

### 3.3 CURVATURE MEASURES

A natural measure of the curvature of the loss landscape is the Hessian of the loss function, defined as $\mathcal{H} \equiv \nabla^2_\theta \mathcal{L}(\theta)$. More precisely, it is the *spectrum* of the Hessian that captures the local geometry of the landscape. However, if the model has $N$ parameters, then $\mathcal{H} \in \mathbb{R}^{N \times N}$, making it intractable to compute or store explicitly for modern networks. Instead, we use three independent summary statistics of the Hessian spectrum, each providing a tractable yet informative proxy for curvature.

We estimate the *maximum eigenvalue* of the Hessian, $\lambda_{\max}(\mathcal{H})$, using the *power iteration method* (see, e.g., Yao et al. (2020)). Crucially, this method requires only Hessian–vector products, which can be computed efficiently via automatic differentiation in $\mathcal{O}(N)$ time. The update rule for the power method is:

$$v^{(n+1)} = \frac{\mathcal{H}v^{(n)}}{\|\mathcal{H}v^{(n)}\|}, \qquad \text{where} \quad \mathcal{H}v = \sum_\beta \frac{\partial^2 \mathcal{L}(\theta)}{\partial\theta_\alpha \partial\theta_\beta} v_\beta. \tag{5}$$

After a few iterations, $v^{(n)}$ converges to the dominant eigenvector, and $\lambda_{\max} \approx \|\mathcal{H}v^{(n)}\|$.

We estimate the *trace* of the Hessian and part of its spectrum using its connection to the Fisher Information Matrix near a minimum. Specifically, when $\theta^\star$ is a local minimum and the model is well-calibrated, the Hessian can be approximated by the Fisher Information Matrix:

$$\mathcal{F}(\theta^\star) \equiv \mathbb{E}_{(x,y)\sim D}\left[s_\theta(x,y)s_\theta^\top(x,y)\right]\Big|_{\theta^\star} \qquad s_\theta(x,y) \equiv \nabla_\theta \log p_\theta(y \mid x) \tag{6}$$

where $s_\theta(x,y)$ is the score. This equivalence is discussed further in the Appendix A.2. From this expression, we can compute the trace of the Fisher—and hence approximate the trace of the Hessian—by summing the diagonal elements of the outer product $s_\theta s_\theta^\top$.

As a third measure, we compute the Fisher matrix on a small random subset of the training dataset of size $E$, and perform singular value decomposition (SVD) on the resulting score matrix, which has shape $N \times (CE)$, where $N$ is the number of parameters and $C$ the number of classes. This procedure efficiently estimates the leading components of the curvature spectrum without requiring full-batch computation or explicit construction of the full Hessian matrix.

#### 3.3.1 A NOTE ON REPARAMETERIZATION

Dinh et al. (2017) showed that symmetries in the architecture of networks allow deep networks to be re-parameterized without changing the function computed by the network. While this observation potentially makes the Hessian a poor tool for studying generalization, we note that when considering SGD optimization dynamics it is still the Hessian and not a reparameterization invariant measure that governs the dynamics of the system. Particularly, for any symmetry $T_\alpha$ that leaves the function computed by the network the same, we have that $\nabla_\alpha L(T_\alpha \theta) = 0$, and so flat directions induced by a symmetry do not induce a gradient.

## 4 RESULTS

### 4.1 ENTROPIC CONFINEMENT

In Figure 2, we show the loss (**C**) and the curvature—quantified by the the trace $\text{Tr}(\mathcal{H})$ (**A**) and the maximum eigenvalue $\lambda_{\max}(\mathcal{H})$ (**B**) of the Hessian—along MEPs connecting different pairs of minima of Wide ResNet-16-4. Interestingly, the loss along the MEP is often lower than at the endpoints. This behavior likely arises because each pivot is pulled downward both by the loss gradient (locally minimizing energy) and by the coupling to the neighboring pivots. This effectively lowers the noise experienced by the system effectively allowing to reach deeper minima. Despite the absence of loss barriers along the MEPs, we observe a sharp rise in curvature along the MEP[1],

---

[1] A small dip is visible near the endpoints in Figure 2(B), where the estimated maximum eigenvalue of the Hessian briefly decreases. We believe this artifact is due to the estimation procedure: computing the Hessian away from an exact minimum introduces a correction proportional to the norm of the gradient. This effect is stronger at the ends of the MEP, where the loss is slightly higher. Interestingly, this dip is not present in the estimates based on the Fisher Information Matrix (panel (**A**) and (**D**) ).

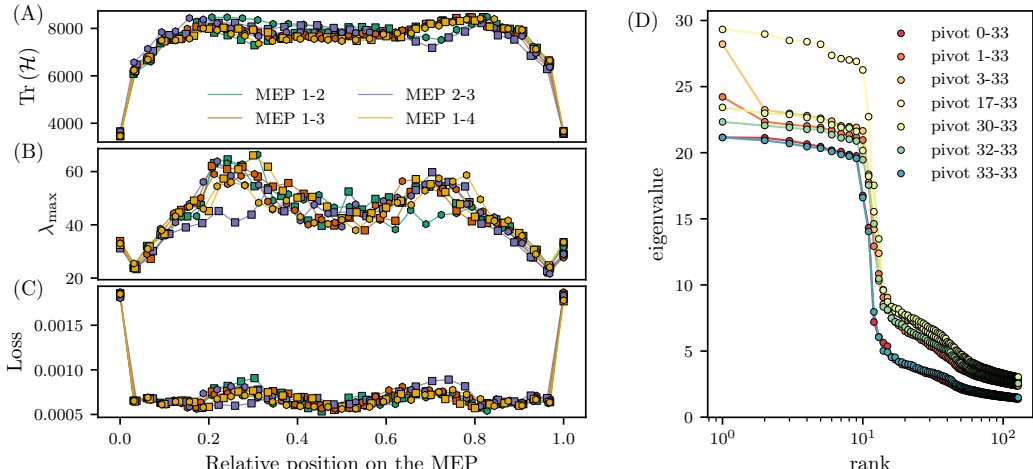

Figure 2: Entropy induces barriers between minima. **(A,B)** Curvature along minimum energy paths (MEPs) connecting different minima, measured via the trace of the Hessian **(A)** and the maximum eigenvalue of the Hessian **(B)**. Numbers indicate distinct minima found via independent training runs, markers indicate pivot points; different colors correspond to different pairs of minima, and marker shapes denote MEPs found via different instantiations of the AutoNEB random seed. **(C)** Cross entropy loss along MEPs connecting different pairs of regular minima. **(D)** Spectrum of the Hessian along MEP 1–2, estimated via singular value decomposition (SVD) of the score matrix computed on $E = 1024$ training examples. As we move into the interior of the MEP, the entire spectrum shifts upward, reflecting an increase in the curvature in all directions along the path.

measured either via $\lambda_{\max}$ or the Hessian trace. The curvature decreases only near the endpoint minima. As argued in Section 2, such variations in curvature generate entropic forces that bias optimization toward flatter regions, even in the absence of explicit loss barriers.

One might argue that the increase in sharpness along the MEP is simply a consequence of the decreasing loss or an effective reduction in learning rate, especially given prior work suggesting a relationship between sharpness and learning rate (Cohen et al., 2021). We argue that this is not the case: while the loss drops between the first and second pivots, it then remains approximately constant along the rest of the MEP. In contrast, both sharpness metrics—maximum eigenvalue and trace of the Hessian—continue to rise. This indicates that the observed increase in curvature is not merely a byproduct of lower loss or implicit regularization, but rather reflects a genuine change in the geometry of the optimization landscape.

### 4.1.1 MEASUREMENT OF ENTROPIC FORCE

To directly observe these entropic effects, we initialize models at specific points along a given MEP and study how stochastic gradient descent pushes them along the path. We use a variant of SGD that projects updates back onto the nearest linear segment of the MEP, ensuring that dynamics remain constrained to the path (see Section A.1 for details). Without this projection, standard SGD causes the models to leave the MEP and wander along other directions in the loss landscape that are not aligned with the MEP. This projected training variant is only used for the experiments in Figures 3, 4 and 8. In this context, to fairly compare experiments that use different learning rates, we plot results against an effective time, defined as the product of the number of optimizer updates and the learning rate, $t_{\mathrm{eff}} = (\text{optimizer updates}) \times \eta$.

As shown in Figure 3(A), when a model is initialized along the MEP, it is pushed back toward the nearest (and relatively flatter) endpoint of the path. Models starting deeper within the MEP take longer to relax to the endpoints. We also note that entropic forces drive the optimization back towards the first pivot despite the fact that the loss actually increases along this path, illustrating a scenario where entropic force is stronger than energetic force. This observation can be understood

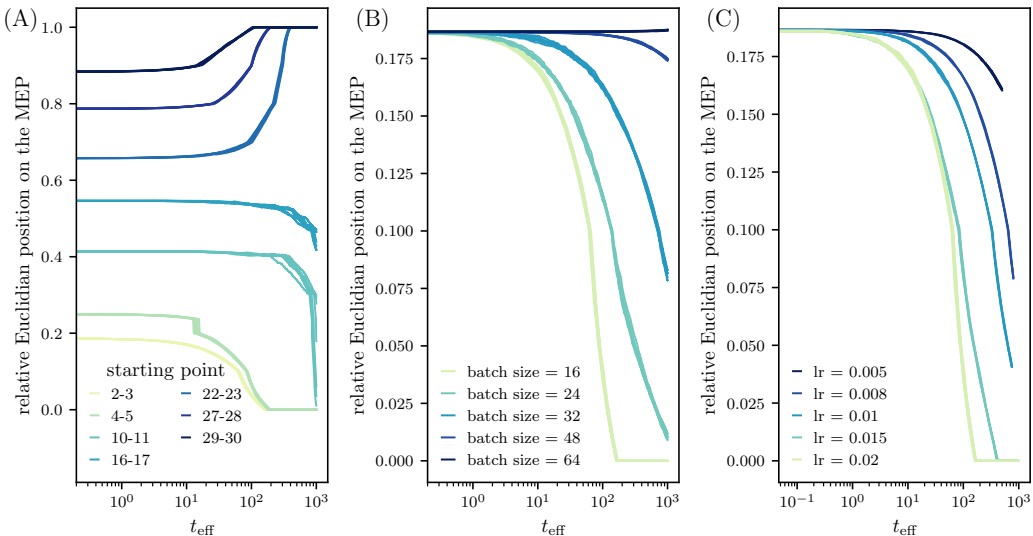

Figure 3: **Relaxation dynamics induced by entropic forces.** (**A**) Relaxation dynamics along the MEP for Vanilla projected SGD (batch size $B = 16$, learning rate $\eta = 0.02$) for models initialized at different points along the MEP (We use MEP 1-2 from Figure 2). The legend shows the two closest pivots to each starting point. Models initialized deeper along the MEP take longer to relax to the endpoint. (**B**, **C**) Models are initialized between the second and third pivots of the MEP, and trained using projected SGD constrained to the path (see Section A.1). The $y$-axis shows the relative *Euclidean* distance along the MEP, where 0 and 1 correspond to the endpoints of the path. The entropic force drives the models back toward the endpoints. (**B**) Models trained with learning rate $\eta = 0.02$ for increasing batch sizes. Relaxation to the endpoint is faster for smaller minibatches, indicating that entropic forces are stronger for smaller batch sizes. (**C**) Models trained with minibatch size 16 for increasing learning rates. Relaxation to the endpoint is faster for larger learning rates, indicating that entropic forces are stronger at higher effective temperatures. Different curves of the same color correspond to different realizations of the SGD noise.

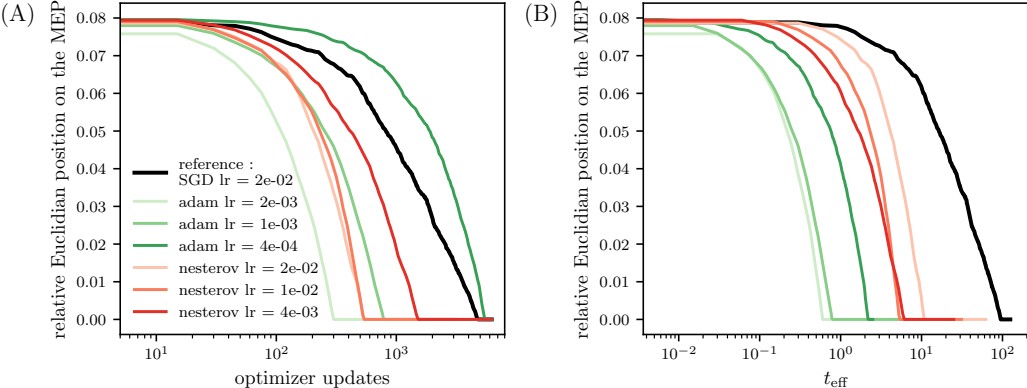

Figure 4: **Relaxation dynamics induced by entropic forces for different optimizers.** Relaxation dynamics along the MEP for projected dynamics using Adam (green) and SGD with Nesterov momentum (red), compared to vanilla SGD (black). We plot the results against the number of updates (A) and the effective time (B). The effect of the entropic forces seems to be more prominent for both Adam and SGD with Nesterov momentum.

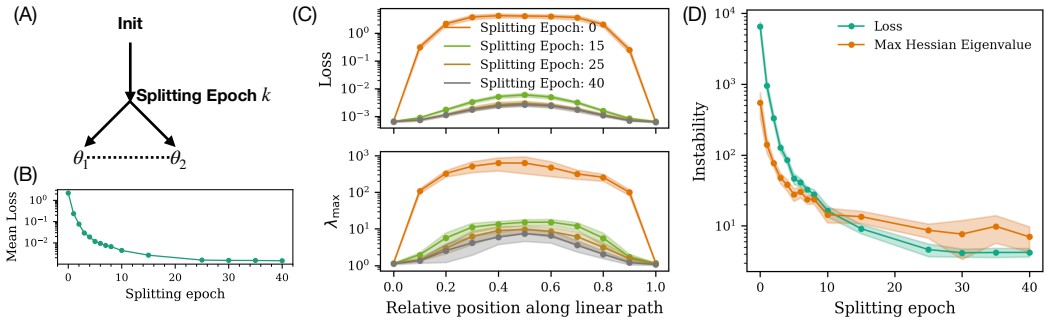

Figure 5: Entropic barriers are relevant later in training. **(A)** Linear mode connectivity schematic (Frankle et al., 2020). We train a network to epoch $k$, then produce two new networks via different data ordering, and measure the loss along a linear path. **(B)** The average loss along such a path goes down as $k$ increases, decreasing rapidly with $k$. **(C)** *Top:* The loss profile along linear paths for various $k$. *Bottom:* The curvature profile, measured by the maximum Hessian eigenvalue, for various $k$. **(D)** We plot the *instability* (The relative change along the path) of the loss and the curvature. For small $k$, the loss exhibits larger instability, while for larger $k$, the curvature exhibits larger instability.

more easily through a statistical physics lens: the noisy dynamics drive the system to minimize not the energy but the *free energy* – the system balances the effects of energy and entropy.

In Figure 3(B) and (C), we show how minibatch size and learning rate affect the dynamics along the MEP. As expected for a genuine entropic force, its strength increases with the effective temperature. Accordingly, relaxation is faster for smaller minibatches, as shown in Figure 3(B), and for larger learning rates (lr), as shown in Figure 3(C). In the Appendix, Figure 8, we show explicitly how the entropic force scales with the batch size.

In Figure 4, we investigate how the choice of optimizer affects the entropic force. We show that (projected) Adam and SGD with momentum both respond *more* strongly to changes in curvature than vanilla SGD. This suggests that the effect of entropic forces may become more important when using adaptive optimizers or using momentum.

The increase of the curvature along the MEP adds nuance to the idea that the loss landscape consists of one large "valley" containing all the parameter configurations with low loss: Although minima in such a valley may be connected energetically, our experiments suggest that such a valley is effectively broken up into disconnected regions by entropic barriers. We emphasize that entropic "barriers" are not barriers in the most literal sense of the word: the model is not dynamically forbidden from crossing such barriers. Rather, the noisy dynamics ensure that crossing an entropic barrier is statistically extremely unlikely. We therefore say that the model is "effectively forbidden" from crossing such a barrier.

## 4.2 LINEAR MODE CONNECTIVITY

Although we have argued that entropic forces separate the low-loss region of parameter space into regions effectively confined by entropic barriers, we have not yet addressed how and when these confined regions are chosen along the course of training. In this section we will take steps towards answering this question through the lens of linear mode connectivity. Following the methods of Frankle et al. (2020), we train $M$ networks with a *shared* data order up until epoch $k$, which we will call the *splitting epoch*. After epoch $k$, each of the $M$ networks sees an *independent* ordering of the data and can then potentially move away from its "siblings," the other $M - 1$ networks. The sibling networks are then trained until convergence. All networks trained in this section use the ResNet-20 architecture (He et al., 2015), unless otherwise noted.

The crucial observation in Frankle et al. (2020) is that once $k$ becomes sufficiently large, the sibling networks become connected by linear paths of low loss, implying that they converge to the same region of parameter space. Interestingly, $k$ does not have to be very large compared to the number of epochs required for convergence before linear mode connectivity is observed. In Figure 5, we

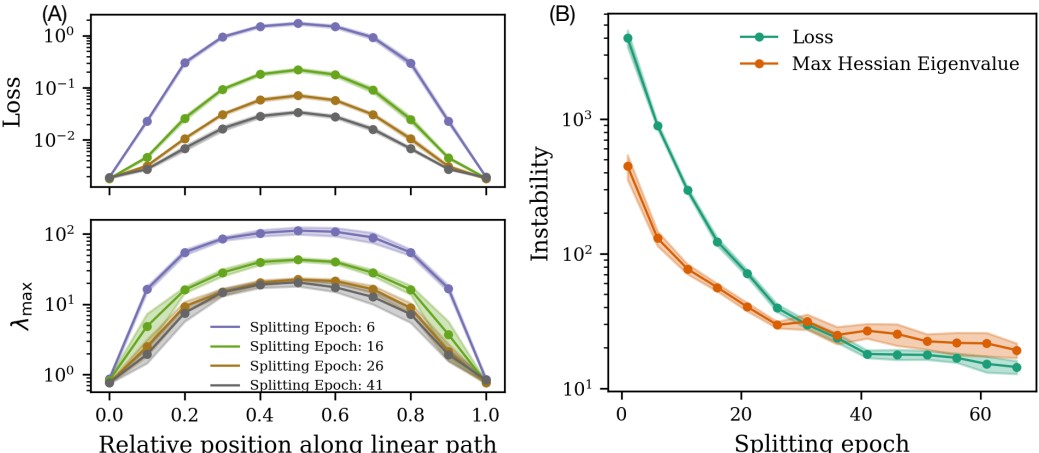

Figure 6: Entropic barrier behavior persists across datasets & architectures. **(A)** The average loss along a linear interpolating path goes down as the splitting epoch $k$ increases, for a ResNet-110 trained on CIFAR-100. **(B)** Entropic barriers become more relevant late in training for a ResNet-110 trained on CIFAR-100.

reproduce these experiments, and measure the curvature along the linear, low-loss paths between converged siblings. Similarly to the nonlinear case (Section 4.1), we see a bump in the curvature along these paths. Additionally, we notice that these entropic barriers persist for larger values of $k$ than their energetic counterparts, implying that entropic forces contribute relatively more to the final stages of the model's localization to a region of parameter space. To see this, we plot the *instability* as a function of $k$ (Figure 5D). The instability measures the relative change, max/min, in the metric (loss or curvature) along the linear path. For small values of $k$, the loss has the larger instability, while for larger values of $k$ the curvature exhibits greater instability.

In Figure 6, we show that this behavior persists across datasets & architectures, and repeat our analysis for a ResNet-110 trained on CIFAR-100. We see similar behaviors in the loss and in the curvature (Figure 6A) across both datasets, emphasizing that these trends are not dataset-specific. We also see similar behavior to Figure 5D for CIFAR-100, where entropic barriers become relatively more important over the course of training (See Figure 6B). In the Appendix Figure 9, we show that a ResNet-20 trained on CIFAR-100 also exhibits similar behavior.

## 5    DISCUSSION

**Entropic Confinement.**    Our results provide new insight into the global geometry of the loss landscape. While prior work has emphasized that minima are often connected by low-loss paths, forming a single broad "valley" of solutions (Garipov et al., 2018; Frankle et al., 2020), our findings reveal that these paths are not flat when entropic forces are taken into consideration (Figure 2). Instead, they exhibit systematic increases in curvature away from their endpoints, producing localized "bumps" in sharpness. This observation refines the valley picture: the basin of low-loss solutions is structured by curvature variations that give rise to entropic barriers.

We show that the forces produced by curvature variations along connecting paths consistently drive optimization dynamics back toward flatter regions near the minima (Figure 3). In particular, models initialized away from a minimum but constrained to remain on the path show persistent drift back toward the endpoint, even though the loss profile is nearly flat. We also observe that smaller batches and larger learning rates accelerate relaxation, showing that the strength of the entropic force depends on the noise level. Entropic forces are not necessarily negligible – we show empirically that they can drive models *up* a loss gradient.

**Entropic Linear Mode Connectivity.**    Our analysis of linear mode connectivity further shows that entropic forces play an important role late in training. As the splitting epoch increases, energetic

barriers along linear paths decrease, but curvature barriers persist for longer into training (Figure 5, Figure 6). This suggests a two-phase picture of training: early dynamics are dominated by energetic forces that drive the model into a low-loss basin, while later on entropic forces become more relevant. Our experiments have important implications for late-time dynamics of deep network training, basin selection, and parameter-space ensembling techniques. In particular, earlier work including Altıntaş et al. (2025), demonstrates that the final basin a network ends up in is highly sensitive to perturbations in the weights, especially early in training. These findings suggest that even small contributions from entropic forces could have an outsized effect on the model's long-term fate.

**Confinement and Generalization.**  Our findings may also provide insight into generalization properties of overparameterized models. Empirically, models trained with SGD tend to find a generalizing solutions and not overfit the data, even after many epochs of training. This occurs *despite* the fact that the loss landscape is energetically flat, raising the question of why optimization dynamics do not diffuse into regions of parameter space that overfit the training data.

We posit that generalizing minima may be effectively disconnected from overfit minima via entropic barriers. Entropic barriers could make paths to such regions effectively inaccessible: even when overfitting solutions are connected to flatter ones by low-loss paths, entropic forces could shield the generalizing solutions by repelling SGD away from regions of parameter space that do not generalize. Our results suggest that this is a promising avenue for future work. In fact, there is evidence that models in similar convex basins of attraction share generalization properties (Juneja et al., 2023).

**Weight-space averaging.**  Our work also provides a new lens through which to view weight-space ensembling techniques. The study of global loss landscape features, such as mode connectivity (Draxler et al., 2018; Frankle et al., 2020), has been crucial in developing methods like Stochastic Weight Averaging (SWA) (Izmailov et al., 2018; Wortsman et al., 2021). Our findings suggest a more nuanced picture of the global landscape: techniques like SWA may be averaging minima that, while energetically connected within a single low-loss valley, may be effectively disconnected by the entropic barriers we observe. This would imply that the SWA solution cannot be easily found by diffusive optimization dynamics at the bottom of a valley in the loss landscape. A valuable avenue for future work would be to analyze the connectivity properties of these averaged minima to better understand how weight-space averaging is able to construct solutions with favorable generalization properties.

**Limitations & Future Work**  We note that there is a large space of low-loss paths connecting minima, and that the methods used here to find such paths (AutoNEB & linear interpolation) introduce a source of bias in the paths we consider. While we acknowledge that this bias could impact the generality of our conclusions, we also observe similar qualitative profiles in the curvature across these two methods, even though they introduce different sources of bias. We believe it is an important and promising direction for future work to investigate how to sample the space of paths in a more principled manner.

## 6  CONCLUSION

We identify a key geometric feature of neural network loss landscapes and its impact on optimization dynamics. Our central finding is that low-loss paths connecting distinct minima consistently exhibit a rise in curvature away their endpoints. We show that this variation, when coupled with the inherent noise of stochastic gradient descent, gives rise to entropic barriers. We demonstrate empirically that these barriers generate effective forces that confine the optimizer to flatter regions near the minima, even when the path is energetically favorable.

Our experiments exploring the curvature along linearly mode-connected networks reveal that the mechanism of entropic confinement is particularly relevant during the later stages of training, shaping the final localization and stability of the learned solution. Our results establish these curvature-induced forces as a key element in understanding the behavior of stochastic optimizers. This geometric perspective offers new insights into how the landscape itself guides the discovery of stable and well-generalizing models, providing a promising direction for future research.

**Ethics Statement.** We do not foresee any direct ethical concerns arising from this work. Our study focuses on better understanding optimization in machine learning, without direct deployment in sensitive application domains. We note that we have used language models to polish the text of this manuscript in places.

**Reproducibility Statement.** We have provided descriptions of all algorithms, models, and experimental setups in the main text and appendix. Training procedures and dataset details are documented to facilitate replication. When the author list is unblinded, we will release our codebase to enable full reproducibility of our results.

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

## A  APPENDIX

### A.1  $k$-STEP PROJECTED SGD

In order to directly measure the effect of entropic forces in a controlled setting, we use a modified version of SGD. Our algorithm deals with two conflicting considerations: First, we would like to limit the scope of our observation to models that lie on a linear path, or more generally models that lie on a MEP. However, we would also like to run optimization in such a way that entropic forces arising from curvature are still relevant to the optimization dynamics. The key observation is that if we were to run SGD on a line in parameter space, projecting back to the line after each optimization step, we would remove the effect of entropic forces, which arise from noisy multi-step optimization dynamics Wei & Schwab (2019). Motivated by this, we propose a natural algorithm that trades off between these two considerations by taking multiple SGD steps between before projecting the parameters back to the liner path (or MEP).

---

**Algorithm 1** $k$-step projected SGD

---

**Input:** A model $f_\theta(x)$, a loss function $L(\theta, x, y)$, an integer $k$, path pivots $\theta_0, \theta_1, \ldots \theta_N$ SGD learning rate $\eta$, SGD batch size $B$.

    **while** not converged **do**

        **for** $i = 1$ to $k$ **do**

            Draw a batch $b \leftarrow \{x_j, y_j\}_{j=1}^B \sim D_{\text{train}}$

            $\theta \leftarrow \theta - \eta \boldsymbol{\nabla}_\theta L(\theta, b)$

        **end for**

        Project $\theta$ onto the closest segment (across $n$) connecting $\theta_n$ and $\theta_{n+1}$.

    **end while**

---

In this way, $k$ trades off the effect of entropic forces (large $k$) vs how close to the linear, low-loss path optimization stays (small $k$). In Figure 3, we use $k = 15$ and run the algorithm along the MEP 1-2. The number of steps plotted on the horizontal axis is the "raw" number of SGD steps – i.e. the number of times the parameter vector of the network was updated.

### A.2  THE FISHER TRICK FOR ESTIMATING THE HESSIAN

Computing the full Hessian of the training loss is intractable for modern neural networks due to both memory and runtime constraints. The Hessian matrix has $\mathcal{O}\left(N_p^2\right)$ parameters, where $N_p$ is the number of parameters of the network, hence even just computing and storing the Hessian matrix is prohibitive. A common workaround is to exploit the equivalence between the Hessian of the loss and the Fisher Information Matrix (FIM) at a minimum.

If, as in the case of image classification, the loss is the negative log-likelihood,

$$\mathcal{L}(\theta) = - \sum_{(x,y) \in D} \log p_\theta(y \mid x).$$

At any parameter vector $\theta$ that minimizes the loss function $\mathcal{L}(\theta)$, we have that $\mathbb{E}_{p_\theta(y|x)} \left[ \log(p_\theta(y|x)) \right] = 0$, taking the derivative of this equation with respect to $\theta$ and using the log-derivative trick we have:

$$\mathbb{E}_{p_\theta(y|x)} \left[ \nabla_\theta \log(p_\theta(y|x)) \right] = \mathbb{E}_{p_\theta(y|x)} \left[ \nabla_\theta^2 p_\theta(y|x) \right] - \mathbb{E}_{p_\theta(y|x)} \left[ \nabla_\theta \log(p_\theta(y|x)) \nabla_\theta \log(p_\theta(y|x)) \right]$$
(7)

Therefore at any minimum $\theta^\star$ of the loss, the Hessian of the loss coincides with the Fisher information matrix $\mathcal{F}(\theta^\star)$,

$$\mathcal{F}(\theta^\star) \equiv \mathbb{E}_{(x,y)\sim D} \left[ \nabla_\theta \log p_\theta(y \mid x) \, \nabla_\theta \log p_\theta(y \mid x)^\top \right] \Big|_{\theta^\star}.$$
(8)

This identity allows us to approximate Hessian eigenvalues using stochastic estimates of the FIM. In practice, the FIM is easier to estimate than the Hessian, since it can be decomposed into a product of low-rank matrices.

## A.3 SUPPLEMENTARY FIGURES

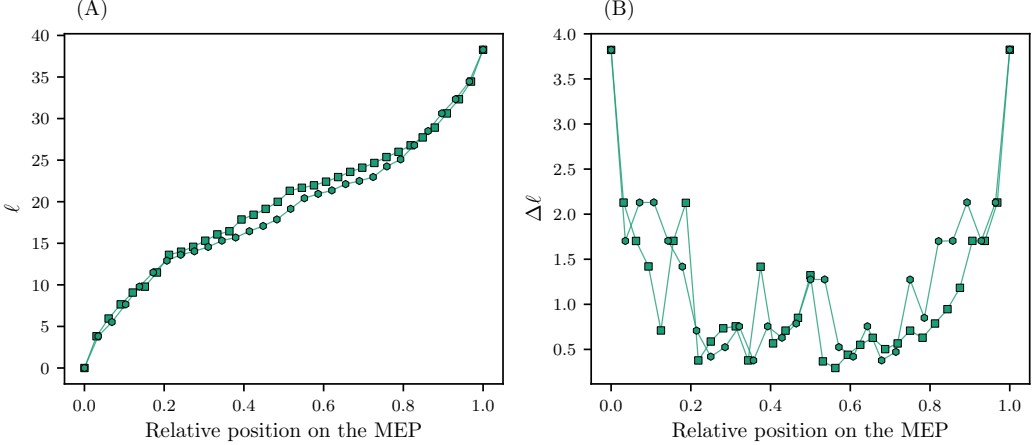

Figure 7: **Properties of MEPs constructed with AutoNEB.** The AutoNEB algorithm updates the positions of pivots without changing the lengths $\ell$ of the segments connecting them, and adds new pivots along segments where the loss is not well-approximated by the linear interpolation, see Draxler et al. (2018). As a result, the Euclidean distance between consecutive pivots is not constant, and pivots tend to be denser near the middle of the MEP. (**A**) Relative Euclidean distance of each pivot measured from the first pivot, illustrating the cumulative distance along the MEP. (**B**) Lengths of individual segments between consecutive pivots, showing that the spacing $\Delta\ell$ is non-uniform. Note that the values in (A) correspond to the cumulative sum of the segment lengths shown in (B).

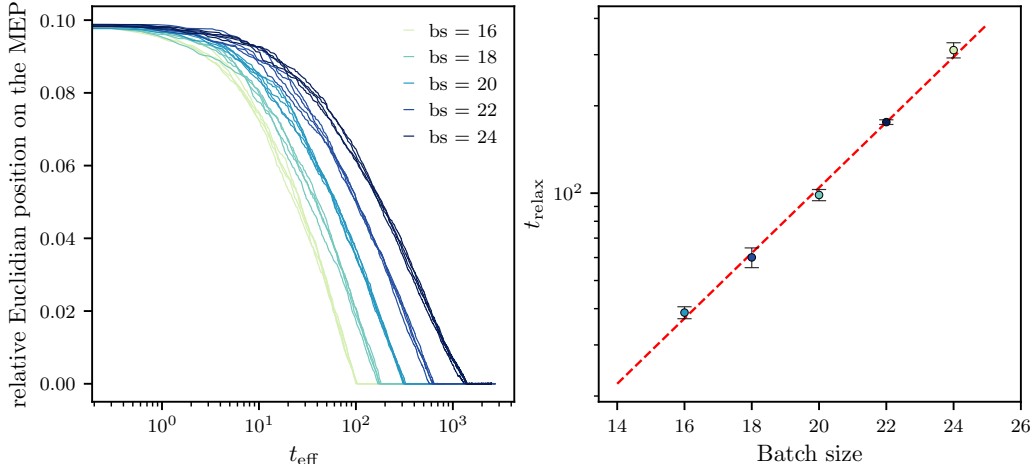

Figure 8: Relaxation dynamics induced by entropic forces. (**A**) Relaxation dynamics along the MEP using Vanilla projected SGD (see Section A.1) with learning rate $\eta = 0.02$ for models initialized at the second pivot of the MEP. Different colors indicate different minibatch sizes, and different curves correspond to different realizations. (**B**) Dependence of the characteristic relaxation time, defined as the time required for the relative distance along the MEP to decrease by a factor of $e$. The relaxation time appears to be well described by a growing exponential.

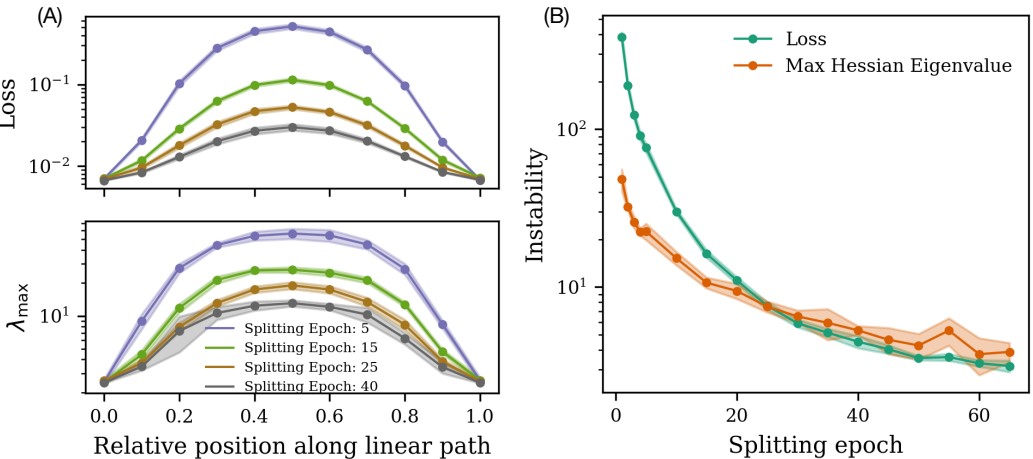

Figure 9: Entropic barrier behavior persists across datasets. (**A**) The average loss along a linear interpolating path goes down as $k$ increases, for a ResNet-20 trained on CIFAR-100. (**B**) Entropic barriers become more relevant late in training for a ResNet-20 trained on CIFAR-100.

