# OpenReview forum: "Entropic Confinement and Mode Connectivity in Overparameterized Neural Networks"
_ICLR.cc/2026/Conference — ICLR 2026 Poster_

### Official Review · Reviewer_zHdP · 2025-10-28

**Soundness:** 3
**Presentation:** 2
**Contribution:** 4
**Rating:** 4
**Confidence:** 4

**Summary:**

The paper describes an entropic mechanism by which stochastic optimization is biased towards flatter minima. The proposed mechanism is that flatter minima take up larger portion of weight space, and since training is noisy, entropic considerations would predict that the posterior distribution of network parameters would be localized in low-curvature zones.  This extends a line of research about the implicit bias of various optimizers, the connection between local curvature and generalization properties, and the connectivity of the low loss manifold of deep networks.

**Strengths:**

I really loved the central idea of the paper. It offers a simple conceptual model and provides concrete and convincing numerical demonstrations. The question it addresses is timely and important, and the answer is at once original, simple and insightful, even if not rigorously proven. The presentation is relatively clear, though I think much of it is wrongly phrased, see below. I think this is an important contribution in this sub field, that will likely generate interesting follow up works.

**Weaknesses:**

Despite the praises above, my final recommedation is a weak reject because I think the manuscript has some fundamental flaws, and proofreading is more than sloppy. However, as I wrote in the Strengths section above, I think the central idea is (probably) valid and interesting and that this manuscript would be a good contribution to the field once the flaws are fixed. I'd be very happy to upgrade my assessment pending the authors response.

1. The main weakness I see is in the framing. The terminology used is at times inaccurate and at times plainly wrong and contains what I think are fundamental mistakes in interpertation. However, the general idea is, I think, valid. The manuscript would be much stronger (and my grade would be higher) if this were corrected. I'll try to as explicit as i can with this point:
  - The main disagreement I have with the framing has to do with how the authors distinguish energy and entropy in DYNAMICAL terms.  For example, the opening statement of Sec. 2 is that "the dynamics of a system are governed not only by energetic forces—derived from gradients of an energy or potential—but also by entropic forces, arising from thermal fluctuations". Similarly, the authors write that "entropy causes optimization to climb up the loss" (line 138) or that "entropic forces dynamically confine models" (line 72) or similar statements scattered throughout the manuscript.
 This is a delicate point: The basic tenet of stat mech is that the Boltzmann distribution arises naturally *even if the underlying dynamics is purely deterministic and energy driven*, the canonical example for which is the micro-canonical ensemble, (sorry for the pun). When the authors write that the system is "dynamIcally confined" to high entropy regions, they use this term exactly in the opposite meaning of what "dynamical confinement" would mean in the stat mech literature -- namely places that have non-vanishing Boltzman weight and that therefore one should be able to probe if dynamics were ergodic. But one cannot probe them because of dynamic (=energetic, not entropic) constraints.
 * Explicitly: in the stat mech jargon, dynamical confinements pertain to *energy* barriers, while entropic confinements pertain to *free energy*. This is a crucial difference, which is central to the authors' main point.
(a trivial textbook example: consider an ideal gas in a container. It is *perfectly dynamically possible* for the gas to be concentrated in one half of the container, but such configurations have very low entropy (high free energy) and therefore it is *thermodynamically* forbidden, while being *dynamically allowed*)

- A similar mistake happens when the authors write that "it is possible for entropic forces to be the stronger than energetic forces, leading to a scenario where entropy causes optimization to climb the loss." (line 138). However, this is generically the case. It is *always true* that "entropic forces" drive dynamics up the energy landscape (driving them down the energy landscape would result in negative temperature which is not strictly forbidden by thermodynamic, but is quite an esoteric condition which I think is irrelevant here). Even in the toy example given by the authors of a valley with a varying curvature, $V=g(x)y^2$, the basic fact that $\langle x^2 \rangle\ne0$ stems from the fact that at finite temperatures (hence finite entropy) the dynamics drive the system up the quadratic well.

I think both these mistakes, and similar ones, stem from the same basic misunderstanding. This framing should be fixed.

2. The main result of the paper is Fig. 2 and the curvature variation along the MEP. However, I suspect that the custom SGD the authors use might affect this result. At the very least, the authors should present how the curvature along MEP looks like if one uses the method of Frankle /Draxler directly, and show how their result changes with $k$ (defined in appendix A1, and, as far as I understand, is confusingly not the same $k$ as in Fig. 4).

3. There are typos, missing references, and two paragraphs are repeated verbatim twice in Sec. 4.2. Appendix A2 terminates in the middle of a sentence (!). This is, frankly, not serious. I know well how submission to these conferences work, but this almost insulting for the reviewers and ACs and all the people who took the time to read your paper seriously and comment on it constructively. The minimum you could do is read it before you submit.

**Questions:**

I wrote my questions in the weaknesses box.

---

> ### Author Response · Authors · 2025-11-21
>
> Thank you for the thoughtful review and the helpful and constructive comments. We apologize for the poor presentation quality and framing and have updated the draft to reflect your feedback.
>
> Specific responses to the listed weaknesses:
>
> 1\. With regards to the framing of entropy as “dynamical,” we agree with and appreciate your comments – as in physical examples, it is not forbidden by the SGD dynamics for the system to reach high-curvature regions of the landscape. Rather, it is simply very unlikely that this will happen. Our original use of the term “dynamical” was intended to refer to the idea that the noise in the SGD dynamics is what creates the entropic force, but we understand how this is confusing. We have updated the paper to instead refer to such situations as “effectively forbidden” in order to emphasize that low-probability events are not strictly forbidden by the equations of motion.
>
> We also update the manuscript to emphasize that “climbing the loss” in this setting corresponds to minimizing the free energy, and we are not claiming anything surprising in the language of statistical physics. Our emphasis of this point is intended to illustrate one of the consequences of free energy minimization that may be surprising to practitioners in machine learning that have spent less time thinking about statistical physics.
>
> 2\. In Figure 2, we directly use the algorithm from Draxler et. al. – the custom projected SGD is only used for the experiments in Figure 3\. We have updated the draft to make this more evident. The main obstacle here is to ensure that the dynamics stay on the MEP – as noted in our responses to other reviewers, running standard SGD from an arbitrary point along the MEP results in the model moving off of the MEP entirely.
>
> 3\. We apologize for the state of the initial manuscript and have gone through it carefully to fix the presentation quality. We take your comments to heart and appreciate your thoughtful consideration nonetheless.

---

> > ### Comment · Reviewer_zHdP · 2025-11-23
> >
> > I think the revised manuscript is in better shape and I upgraded my overall assessment.

---

> > > ### Author Response · Authors · 2025-12-03
> > >
> > > Thank you! We really appreciate your feedback and thoughtful consideration of the paper. We’re pleased our updated manuscript addressed your questions and appreciate your significant score increase.

---

### Official Review · Reviewer_ukzS · 2025-10-29

**Soundness:** 2
**Presentation:** 3
**Contribution:** 3
**Rating:** 4
**Confidence:** 4

**Summary:**

This paper explores the paradox that neural network minima are connected by low-loss paths, but SGD confines itself to individual solutions. The authors propose that entropic barriers, following from the interplay between curvature of the landscape and noise in the optimization, explain the confinement. They run experiments on ResNet architectures trained on CIFAR-10 to demonstrate that the curvature systematically increases along low energy paths between minima. This creates effective forces that bias SGD back to the low curvature endpoints. They show that the magnitude of these forces inversely scales with batch size and argue that these forces persist longer than energetic barriers, becoming dominant in later stages of training.

**Strengths:**

- The paper's motivation is well presented and the introduction/literature review clearly set up the problem.
- The paper connects two well-known phenomena: mode connectivity and the low curvature bias of gradient-based optimizers. These two nicely merge in the paper's analysis, giving an intuitive and convincing picture that the proposed explanation is actually behind the  phenomenon observed.
- I think that the experimental setups are interesting and well-designed. I particularly like the idea of initializing in the middle of the path and looking at the relaxation when performing gradient descent projected on such curve.

**Weaknesses:**

1. My main concern with this work is that, while the ideas are very interesting, the experimental validation is too limited in scope to fully validate it. Given that the paper makes the strong claim that these energy barriers are a "univeral" phenomenon, the experiments should definitely be performed on more than two architectures (Wide ResNet-16-4 and ResNet-20) and one dataset (CIFAR 10) like the paper currently does.

2. Overall, the feeling I get from reading the paper is that a lot of space is used for things that are not central to the argument, like the details about curvature measures and their computation (which could have been an appendix) and the toy model that, in its current form, doesn't really help in conveying the idea (more below).  This space, in my opinion, would have been better spent by expanding the experimental setup.

3. It is not clear how the analytical toy model helps in understanding the phenomenon. First, the derivation feels disconnected. It is not clear how the authors go from Eq. 2 to Eq. 3. Expanding in an appendix would help but the main text section should still be adjusted to be more readable. Furthermore, I don't understand what we are getting from the analytical part of toy model described in page 2. Unless I am missing something, the main message i.e. "regions with smaller curvature $g(y)$ (flatter directions in $x$) contribute larger entropy and are statistically favored, even if the original energy $V(x, y)$ is minimized elsewhere" doesn't seem to follow directly from the analytics. It is however clear enough from Fig. 1B.

4. The claim that "entropic barriers persist longer than energy barriers", which is based on Figure 4D seems a bit overreaching given that it results from only one set of experiment.

### Minor weaknesses

1. I think that the sentence at L249, stating that "symmetries do not have any affect on optimization dynamics." is incorrect. There are plenty of works that study the relationship between symmetries and optimization dynamics. To mention only a few, see [1-3].
2. "delta correlated Gaussian noise" undefined at L091
3. In eq. 2 $g_y$ is the derivative of $g$ w.r.t. $y$?
4. Paragraph duplicated at line 357
5. Figure 2 caption has a ??
6. Page 4 has score(?) after equation 5
7. Missing space at line 56



- [1]  Du, Simon S., Wei Hu, and Jason D. Lee. "Algorithmic regularization in learning deep homogeneous models: Layers are automatically balanced." Advances in neural information processing systems 31 (2018).
- [2] Kunin, Daniel, et al. "Neural mechanics: Symmetry and broken conservation laws in deep learning dynamics." arXiv preprint arXiv:2012.04728 (2020).
- [3] Zhao, Bo, et al. "Symmetries, flat minima, and the conserved quantities of gradient flow." arXiv preprint arXiv:2210.17216 (2022).

**Questions:**

1. Why does starting from 0.7 relaxes the minimum in relative position 0 in Fig.3A? Does it converge to the one in position 1 for higher initializations? It would be interesting to have a more complete profile of what's going on for different starting points (see comments above).
2. What do you think would happen if you initialized in the middle of a path and then perform SGD without projecting to the path?
3. Do the curvature bumps persist when regularization is added to the loss?

---

> ### Author Response · Authors · 2025-11-21
>
> Thank you for the helpful suggestions and constructive comments. We have significantly updated the paper along a number of axes, including fixing all typos (thank you\!) in response to your feedback.
>
> To address the weaknesses mentioned:
>
> 1. We run additional experiments on CIFAR-100 with two different network architectures (Resnet-20 \+ additional architecture in the coming days\!), expand the mode connectivity analysis in Figure 5, and analyze the effect of the learning rate on the entropic force in Figure 3.
> 2. Thank you for pointing out that our presentation could be improved in these ways. We streamline the presentation where possible and better motivate the toy model (see next point).
> 3. We clarify the experimental setup and elaborate on the motivation of the toy model and how it sheds conceptual light on the central ideas of the paper.
> 4. As mentioned in point 1, we run additional experiments on CIFAR-100 with two architectures to strengthen this claim.
>
>
> To address your questions:
>
> 1. We apologize for the confusion and our lack of clarity here. In the original figure, 0.7 meant “70% of the way between the first and second pivots of the path.” Since the first pivot is at relative position 0, we should expect to converge to this relative position. We update the figure, Figure 3, to now use the same relative position coordinates as Figure 2, and also show new experiments initialized across a broader range of positions along the path, some of which indeed converge to relative position 1\.
> 2. When initializing in the middle of a path and then performing standard SGD, we observe that the model drifts away from the initial position, but also drifts off the path entirely. (These experiments were actually the main motivation for our projected-SGD algorithm). We added a comment mentioning this in the manuscript.
> 3. The experiments presented indeed exhibit bumps in the curvature even in the presence of weight decay. We now make sure to emphasize this point in the manuscript.

---

> > ### Comment · Reviewer_ukzS · 2025-11-24
> >
> > I thank the authors for their response and update to the paper.
> > I have a few extra questions/suggestions:
> > - "instability" is defined at L425 as the "fractional change in the metric (loss or curvature) along the linear path". If it is a fraction, how can the y-axis in Fig.4D and Fig.5D display values above 1? Either the definition or the plotting are wrong.
> > - I like the newly updated Fig. 3 but the positioning should be changed to not have it occupy a whole page by itself.
> > - I think the plot displaying the relationship between relaxation time and batch size was interesting and should be put back into the paper, maybe in the appendix.

---

> > > ### Author Response · Authors · 2025-12-03
> > >
> > > Thank you for the further suggestions\!
> > >
> > > * We update the draft to clarify the definition of instability, defined as the maximum value of the metric (loss/curvature) divided by the minimum. This quantity can attain values larger than 1\. We update the language to use the term “relative” change instead of fractional change to clarify this point.
> > > * We also rework the formatting so that Fig. 3 does not take an entire page.
> > > * We improve and re-introduce the plot investigating scaling of entropic forces with batch size into the Appendix.

---

### Official Review · Reviewer_im1k · 2025-11-01

**Soundness:** 3
**Presentation:** 2
**Contribution:** 3
**Rating:** 6
**Confidence:** 3

**Summary:**

This paper provides a dual view on why neural network optimization remains localized near certain minima despite low-loss connecting paths between different solutions. The authors investigate the training dynamics using the AutoNEB algorithm (Draxler) and show that while energetic barriers (arising from the loss) remain low along these paths, entropic barriers emerge from the interaction between curvature variations and stochastic gradient descent noise, manifested as a "curvature bump" away from minima. They demonstrate that these curvature-induced entropic barriers persist longer than traditional energetic barriers and play a key role in determining which regions of parameter space are dynamically accessible during training.

**Strengths:**

- The paper provides an interesting and fresh perspective on how optimization is biased towards certain solutions, driving interesting connections to the implicit bias of SGD (though the relationship to the literature could be expanded here)
- The paper is sound. They use trace and max. Eigenvalue of the Hessian and using the SVD of the Fisher, providing robust evidence that curvature systematically increases along connecting paths.
- The observations relating to the spawning experiments in Frankle are particularly insightful. I would encourage the authors to expand on this connection and engage deeply with the literature on the chaotic early phase of neural networks (e.g. Fort et al, 2020, Altıntaş et al, 2025). The controlled noise setting in Altıntaş could be an interesting addition to this work.

**Weaknesses:**

- While the core findings are convincing, the work would benefit significantly from additional ablation studies. Given that the experimental setting (ResNet-20, Wide ResNet-16-4) is not prohibitively expensive, several natural extensions would strengthen the paper:
- How do entropic barriers manifest when learning rates are reduced? Do they become weaker (less noise)?
- Algorithms like K-FAC, Shampoo, or natural gradient descent explicitly account for curvature. Do these methods navigate entropic barriers differently?
- I think an experiment in at least Cifar-100 scale would be needed.

**Questions:**

- I believe the pivots in the AutoNEB procedure are initialized equidistantly along a linear interpolation between minima (following Draxler et al.), but the subsequent relaxation and potential addition/removal of pivots means they no longer maintain equal spacing. Could the authors clarify: how does the relative position translate into the relative metric position between the two points?
- The existence of entropic barriers along low-loss paths raises intriguing questions about weight averaging techniques like Stochastic Weight Averaging (SWA), model soups, and linear mode connectivity ensembles. If curvature increases along connecting paths between minima, what does this imply for intermediate points obtained by weight averaging?

Minor comments

- The phrase "bottom of the loss landscape" sounds somewhat unnatural, you could consider using more precise terminology like "low-loss regions," "basins of attraction,"
- Missing refs on Lines 208, 233
- First two paragraphs of 4.2 repeated twice verbatim
- MEP 1-2 could be clarified in the figure

---

> ### Author Response · Authors · 2025-11-21
>
> Thank you for your helpful comments and suggestions to improve the paper. We particularly appreciate your pointing out connections to Fort et al, 2020 and Altıntaş et al, 2025, and we update the manuscript to emphasize how Altıntaş et al, 2025 in particular shows how small perturbations during the early phase of training can lead to a different final basin for the network.
>
> In response to the listed weaknesses, we perform several ablations:
>
> * We perform mode connectivity experiments on the CIFAR-100 dataset using two network architectures (Resnet-20 \+ another figure to be added in the coming days\!)
> * We investigate the effect of learning rate on the scale of the entropic force (Figure 3).
> * We investigate the differences between SGD and adaptive optimizers (Adam), and show that Adam responds *more* strongly to changes in curvature (Figure to be added in the coming days\!).
> * We add a plot that shows how the Euclidean distance corresponds to relative distance along the MEP.
>
> In response to your questions:
>
> * To clarify this point, we add a figure to the appendix illustrating the correspondence between relative position and the euclidean distance along the path.
> * We agree that our results may have important implications for techniques that leverage weight averaging, including SWA and model soups. We reserve these interesting questions for follow-up work as they get into questions of generalization which are beyond the scope of this manuscript.

---

> > ### Author Response · Authors · 2025-12-03
> >
> > Just a brief update that we have now added the additional experiments we promised above, specifically: experiments on a new dataset (CIFAR-100), new architecture (ResNet-110), and an adaptive optimizer (Adam).

---

### Official Review · Reviewer_rx5A · 2025-11-01

**Soundness:** 2
**Presentation:** 1
**Contribution:** 3
**Rating:** 4
**Confidence:** 3

**Summary:**

Minibatch training introduces stochasticity, which is also modulated by the discrete step size.
This paper studies so-called *entropic forces*, an implicit bias that progressively nudges optimization toward flatter regions of the loss landscape, strengthening as update noise increases.
Seen through the lens of mode connectivity, this driving force is proposed as an dynamic barrier between disconnected minima regions.
Indeed, by sampling low-loss paths between minima with AutoNEB, they observe that curvature rises away from the endpoints.
When models are initialized slightly off these endpoints, path constrained dynamics pull them back, revealing a curvature-driven “gatekeeping” effect that intensifies with smaller batch sizes.
Reusing a setup of linear mode connectivity, the authors further argue that the relative strength of the entropic force grows in the later stages with respect to the loss.

**Strengths:**

This paper presents an interesting study curvature-induced biases in neural network optimization.
It offers valuable insights into how stochasticity combined with geometry can influence the parameter dynamics, mostly towards the end of training when the loss' role is lessened.
The paper contributes to explain why certain solutions, though equivalent from the point of view of the train loss, are preferred and therefore challenges our understanding of the implicit biases of optimizers.
The results could have implications for late-stage optimization dynamics, low-loss manifolds of solutions, stability within the zero-loss regime, and connections to phenomena such as double descent or grokking.
An original point of the paper lies in the fine-grained, directional analysis of local curvature effects beyond more general arguments given in previous literature (e.g. Edge of Stability, catapult).
The work also raises an interesting question on overfitting, that could be countered by such entropic forces and hence foster better generalization.
In that sense it might also shed more light on why flatness measures have been successful despite not being principled.

**Weaknesses:**

While the paper is conceptually interesting and seems technically sound, it suffers from awkward or overly elliptical phrasing, despite no apparent space constraints (see questions and minors below for specifics).
This gives the impression that the manuscript was not thoroughly reread, which also somewhat undermines confidence in the results.

The paper sometimes seems to overreach:
- line 155: The reference to “the constant loss non-linear path” overstates uniqueness. In addition, identifying paths with the Auto NEB algorithm means incorporating any bias towards particular properties these paths might have, and this is not discussed. (see my question about it below)
- line 320 & 406: the claim that the entropic forces are responsible for the localization of the parameter when the *splitting epoch* is high appears is not fully supported.
Figure 4 only suggests a relative increase in the importance of entropic forces, not that they overtake loss dynamics. A clearer analysis of the trade-off between energetic and entropic components would strengthen this point.

Interpretation of Fig 2 c.: while earlier works have shown that intermediate models can outperform endpoints (e.g. Model Fusion via Optimal Transport by Singh and Jaggi or Git Re-Basin: Merging Models modulo Permutation Symmetries by Ainsworth et al), the abrupt loss drop in Figure 2(c) seems unusual. The explanation given by the authors line 260 is not very convincing as it should also applies in earlier works published since the work by Draxler et al. (see also my related question below)

Experimental details are reported but code is currently not provided.

Limitations include:
1. not discussing quantitatively the tradeoff between entropic forces and loss which makes it difficult to quantify the effect of entropic forces on a typical training
1. while the batchsize point of view is interesting, it would have been great to also investigate the learning rate since it is mentioned as the other source of stochasticity in section 2
1. while making broader connections and mentioning potential applications, the paper focuses only on the mode connectivity/linear mode connectivity setup

## Minor and additional feedback
- duplicated paragraphs in 4.
- line 621 trimmed or spurious full stop before "in practice" ?
- line 233 equation ??
- fig 2: plots inverted w.r.t. caption
- line 93: $\eta$ and $B$ are not introduced
- line 100: isn't there a missing $T$ in the variance formula ?
- line 70: when the loss is near $0$ is a bit imprecise, is it "when the loss is near zero along the path of solutions"
- line 100 and equation 2: I think it would be better to adopt a consistent notation for the variance
- the whole toy model example is a bit too elliptical (also while $g_y$ is understandable is it not clearly defined and adds to the reader disambiguation effort)
- line 208 missing link (?)
- The sign of $V_{eff}$ might be wrong in equation $3$ since it leads to $P(y)\propto \text{exp(ln }g(y))$ and so the dynamics would be attracted towards high curvature region instead of repulsed.
- $m$ of figure 3B is called $\rho$ in the main text
- abstract: "yet optimization dynamics remain confined to one solution" different parameters can be sampled at the end of training so this is a bit unclear
- line 130: 'entropic' is spelled 'entorpic'
- line 249: 'affect' is written instead of 'effect'
- line 596: "on along" is redundant

**Questions:**

1. **Subsection 4.1** you mention that the loss drop may result from the elastic coupling between pivots. Could you elaborate on how this coupling contributes to a further decrease in loss?
1. **Figure 3 setup**: what behavior do you observe if the parameters are not projected back onto the MEP during training? This question seems important to assess the scope of entropic forces in a real training process for example.
1. **Figure 2 curvature plateau** – Around the midpoint of the MEP, curvature metrics appear nearly constant, with even a local minimum of $\lambda_{\text{max}}$ on Fig 2B. . Should we interpret this as a region where entropic forces vanish, leaving the parameters unbiased toward the endpoints? Have you tested initializations in this region?
1. **AutoNEB bias** AutoNEB minimizes both loss and path length: "the spring energy grows quadratically with the total length of the path" (quote from Drawler et al). Could this introduce a bias toward paths that wrap around high-loss regions or exhibit higher curvature? How sensitive do you believe your findings are to the specific properties of the AutoNEB algorithm?

---

> ### Author Response · Authors · 2025-11-21
>
> Thank you for the helpful and constructive review and thought you put into reading our work. We apologize for the presentation and have updated the text to remove elliptical phrasing, fixed your various minor feedbacks (thank you\!), and significantly reworked the presentation for clarity. In particular:
>
> * We are now careful not to overstate uniqueness around line 155 of the original manuscript. We now note that there is a large space of low-loss paths connecting two minima, that we are unaware of tractable methods for sampling this space without bias, and that in our work we study two well-established methods of finding such paths: AutoNEB and linear interpolation (for the mode connectivity experiments).
> * We now clarify explicitly that the mode connectivity experiments show only that the entropic force becomes more relevant as training progresses and do not claim it is responsible for localization. We also added new experiments showing how the entropic force scales with learning rate
>
> Regarding the drop in loss near the endpoints in Figure 2C, we would like to emphasize that this drop is imperceptible when plotted on the same axes as the loss along a linear interpolation (as in e.g. Draxler et. al. Figure 1 and Figure 4, which uses a linear axis from 0 to 4 or 2, respectively).
>
> Proper quantification of the magnitude of the entropic force compared to the gradient of the loss is challenging and depends on how far training has progressed, and our aim here is primarily to establish that entropic confinement as a phenomenon actually occurs in low-loss regions of parameter space, where energetic forces are negligible.
>
> Responses to your specific questions:
>
> 1. In a low-loss basin, a reason the “final” loss value may be higher is due to stochastic fluctuations around a minimum arising from the minibatch noise. The coupling between pivots serves to restrict the movement of each network, at least along a certain direction, and so it effectively reduces the magnitude of these fluctuations. We also emphasize that the scale of the difference is small (see response to question 4 as well). We update the manuscript to better explain this point.
> 2. When initializing in the middle of a path and then performing standard SGD, we observe that the model drifts away from the initial position, and it also drifts off the MEP entirely. (These experiments were in fact the main motivation for our projected-SGD algorithm). We added a comment on this point in the manuscript.
> 3. We update Figure 3 to test initializations along the entire MEP, and we now show that the relaxation towards the endpoint for models initialized deeper in the MEP is slower.
> 4. It is not clear to us that AutoNEB directly minimizes the path length – the quote you mention from Draxler et. al. refers to the “standard” NEB algorithm without pivot insertion, and indeed Draxler et. al. eventually find that an infinite spring constant works best for their task (we replicate this choice). In fact, AutoNEB was designed so that  when the true loss computed along the segment composing the MEP is much larger than its linear interpolation, a new pivot is added, so the path may grow in length with no penalty. Nonetheless, we take your point that AutoNEB may introduce a bias towards specific types of paths, and we update the text to discuss this point more comprehensively. We note, however, that the linear paths we consider in the mode connectivity experiments also exhibit a similar rise in the curvature despite not arising from the AutoNEB algorithm.

---

> ### Comment · Reviewer_rx5A · 2025-11-26
>
> Thank you for the responses and the additional experiments.
>
> I agree with reviewer ukzS that Figure 3 should ideally not appear alone on a page. You might also consider reversing the colormap of the right-hand plot so that higher noise (learning rate) has a consistent visual meaning with the middle plot.
>
> The explanation of how coupling could help reduce the loss is still not fully convincing, though this is not a major issue. I agree that AutoNEB may be less sensitive to high-curvature regions thanks to pivot insertion, and I appreciate the clarifications added to the manuscript.
>
> While I do not doubt the existence of entropic forces, the paper would have greater impact if it could demonstrate a “natural” regime in which energetic forces are negligible and the dynamics are primarily entropic.
> As now stated in the paper, figure 4 and 5 only show that the relative importance can evolve in favor of entropic forces along training.
> Constrained optimization on a 1D path pushes the parameter in the expected direction, but in the unconstrained case the trajectory leaves that path, suggesting that entropic forces may not play the same role (though this does not rule out any effect).
> The last argument that entropic forces can drive gradient ascent is intriguing but not yet sufficiently robust.
> Still, these forces have a clear manifestation in the toy models (Figure 1), and determining whether they play a meaningful role in deep learning practice remains a nontrivial question.
> Overall, the authors have satisfactorily addressed several of my earlier concerns and updated the manuscript accordingly. Even though the practical impact of entropic forces is not conclusively demonstrated, I will raise my score because I think the current version already has sufficient value.

---

> > ### Author Response · Authors · 2025-12-03
> >
> > Thank you for your helpful comments and for updating your review and score! We reworked the manuscript so that Figure 3 does not appear alone on a page. We also agree that sampling low-loss paths with AutoNEB introduces a source of bias, and we make sure to emphasize this point in the paper.
> >
> > We also agree that demonstrating the effect of entropic forces during “natural” training (as opposed to the projective variant of SGD we consider) would be interesting, though it would require a new experimental framework, and we believe that this is a promising avenue for future research. We emphasize that the loss landscape has many flat modes along the paths considered, and so unconstrained optimization is very likely to explore directions orthogonal to the MEPs we find.
> >
> > We’re glad you find the current version already has sufficient value, and we are excited that our results point to interesting future work.

---

### Author Response · Authors · 2025-12-03
**Comment to New Area Chair**

To the Area Chair:

Thank you for your time and consideration of our work, especially in light of the unusual circumstances. We would like to state clearly that we did not know and still do not know the identities of the reviewers, and did not communicate with them outside of OpenReview. We would like to summarize the prior (very helpful\!) discussion and highlight the ways in which the manuscript has improved since submission. We thank the reviewers for their careful feedback; the discussion was highly constructive and their suggestions have substantially improved the paper.

* A primary consistent theme was a need to improve the clarity & editing of the original manuscript. We significantly updated the manuscript following this feedback and are pleased that several of the reviewers think the presentation is now much clearer.
* We performed several new experiments and added new figures to the paper:
  * We performed new experiments on CIFAR-100, as well as on new architectures (ResNet-110).
  * We investigated the dependence of entropic forces on learning rate & optimizer, including Adam as an adaptive optimizer.
  * We expanded the scope of the experiments measuring entropic force, initializing models along a greater range of initial relative positions.
* We clarified the framing of entropic forces, connecting better with the statistical physics literature.

We would like to mention that in light of this discussion, several of the reviewers raised their scores from 4/6/4/4 to 6/6/4/8 for reviewers rx5A/im1k/ukzS/zHdP before the rebuttal period was closed early. After the closing of the rebuttal period, we updated the manuscript again to respond to all of the reviewers’ remaining questions, with responses to each point below. We believe the paper is further improved beyond when these latest scores were submitted.

We thank you again for your time and careful consideration, and appreciate your flexibility in light of the unusual review process this year\!

---

### Meta-Review · Area_Chair_D3ZG · 2025-12-31

**Summary:**

This paper identifies entropic barriers that constrain the optimization dynamics into a single convex basin. The idea is that low-loss paths are not explored because there the curvature rises producing forces that bias the dynamics back to the basin. To demonstrate this experimentally, the authors sample low-loss paths between minima using the AutoNEB algorithm introduced by Draxler et al. (2018): along those paths, the interaction between curvature variations and the noise coming from stochastic gradient descent create entropic barriers that bias SGD.

All the 4 reviewers have praised the interesting and original idea at the basis of the manuscript, which provides a remarkable connection between mode connectivity and low-curvature bias of gradient descent. A recurring issue in all the reviews is the presentation with several parts that are unclear (or just plain wrong) and plenty typos/imprecisions. To address this, the authors have provided a thoughtful revision, which also contains additional experiments (another issue pointed out in multiple reviews). As such, in my opinion, most of the concerns of the reviewers have been solved and I am happy to recommend the paper to be accepted.

**Reviewer Concerns:**

The authors have successfully addressed the issues concerning the framing (reviewer zHdP), the poor presentation (all reviewers) and absence of additional ablations (reviewers im1k, ukzS). Some of the concerns raised by reviewer rx5A have not been completely solved (explanation of how coupling could help reduce the loss; demonstration of a 'natural' regime in which energetic forces are negligible and the dynamics is primarily entropic). However, these outstanding issues are not significant enough to preclude the acceptance of the paper.

**Reviewer Scores:**

Reviewer scores were already raised before the discussion period had to come to an abrupt end and, in my opinion, some form of consensus towards acceptance could have been reached if the reviewers had been able to participate fully in the discussion.

---

### Decision · Program_Chairs · 2026-01-26

Accept (Poster)